# Phosphorylation of β-arrestin2 at Thr[383] by MEK underlies β-arrestin-dependent activation of Erk1/2 by GPCRs

Elisabeth Cassier[1,2,3], Nathalie Gallay[4,5,6], Thomas Bourquard[4,5,6], Sylvie Claeysen[1,2,3], Joël Bockaert[1,2,3], Pascale Crépieux[4,5,6], Anne Poupon[4,5,6], Eric Reiter[4,5,6], Philippe Marin[1,2,3]*, Franck Vandermoere[1,2,3]*

[1]CNRS, UMR-5203, Institut de Génomique Fonctionnelle, Montpellier, France; [2]INSERM, U1191, Montpellier, France; [3]Université de Montpellier, Montpellier, France; [4]INRA, UMR85, Unité Physiologie de la Reproduction et des Comportements, Nouzilly, France; [5]CNRS, UMR7247, Nouzilly, France; [6]Université François Rabelais, Tours, France

**Abstract** In addition to their role in desensitization and internalization of G protein-coupled receptors (GPCRs), β-arrestins are essential scaffolds linking GPCRs to Erk1/2 signaling. However, their role in GPCR-operated Erk1/2 activation differs between GPCRs and the underlying mechanism remains poorly characterized. Here, we show that activation of serotonin 5-HT$_{2C}$ receptors, which engage Erk1/2 pathway via a β-arrestin-dependent mechanism, promotes MEK-dependent β-arrestin2 phosphorylation at Thr[383], a necessary step for Erk recruitment to the receptor/β-arrestin complex and Erk activation. Likewise, Thr[383] phosphorylation is involved in β-arrestin-dependent Erk1/2 stimulation elicited by other GPCRs such as β$_2$-adrenergic, FSH and CXCR4 receptors, but does not affect the β-arrestin-independent Erk1/2 activation by 5-HT$_4$ receptor. Collectively, these data show that β-arrestin2 phosphorylation at Thr[383] underlies β-arrestin-dependent Erk1/2 activation by GPCRs.

*For correspondence: philippe. marin@igf.cnrs.fr (PM); franck. vandermoere@igf.cnrs.fr (FV)

**Competing interests:** The authors declare that no competing interests exist.

## Introduction

Arrestins were initially named on the basis of their ability to turn-off the coupling of G protein-coupled receptors (GPCRs) to G proteins and thereby inhibit G protein-dependent GPCR signaling. Over the last decade, it has become evident that β-arrestins are also important signal transducers and that a number of biological functions exerted by GPCRs are mediated by β-arrestin-dependent signaling (*Shenoy and Lefkowitz, 2011*; *Lefkowitz and Shenoy, 2005*; *Latapy and Beaulieu, 2013*; *Beaulieu et al., 2007*). β-arrestins act as multifunctional scaffolds that recruit multiple signaling molecules, such as the core Extracellular signal-regulated kinase Erk1/2 signaling module, composed of Raf1, MAP kinase kinase MEK1 and MAP kinases Erk1/2, a process leading to Erk1/2 activation (*Reiter and Lefkowitz, 2006*; *DeFea et al., 2000*; *Luttrell et al., 2001*). Typically, most GPCRs induce temporally distinct and spatially segregated G protein-dependent and β-arrestin-dependent activation of Erk: G protein-mediated Erk activation is rapid and transient, reaching a maximal level within a few minutes and is followed by translocation of activated Erk into the nucleus to promote gene transcription and cell proliferation. In contrast, β-arrestin-mediated Erk1/2 activation is slower in onset, requiring 5–10 min to reach maximal level, persists more than 1 hr and mainly occurs in the cytosol (*Reiter and Lefkowitz, 2006*; *Ahn et al., 2004a*; *Shenoy et al., 2006*). However, Erk1/2 signaling elicited by certain GPCRs does not fit this classic bimodal activation pattern: for example, the activation of Erk1/2 pathway by 5HT$_{2C}$ receptors is entirely dependent on β-arrestins

(*Labasque et al., 2008*), whereas the engagement of this pathway by the 5-HT$_4$ receptor does not require $\beta$-arrestin recruitment (*Barthet et al., 2007*). In addition, some GPCRs such as $\mu$-opioid receptor induce $\beta$-arrestin-dependent activation of Erk1/2 that translocate to the nucleus (*Zheng et al., 2008*).

Accumulating evidence indicates that $\beta$-arrestin recruitment to GPCRs and engagement of $\beta$-arrestin-mediated signaling depend on both receptor conformational state and a complex pattern of receptor phosphorylation elicited by GPCR kinases (GRKs) (*Reiter and Lefkowitz, 2006*; *Shukla et al., 2011*). This phosphorylation barcode is translated into specific $\beta$-arrestin conformations that dictate selective signaling and the nature of $\beta$-arrestin intracellular functions. Another mechanism that might underlie $\beta$-arrestin dependency of receptor-operated signaling is the phosphorylation of $\beta$-arrestins themselves. Several phosphorylated residues have been identified in the $\beta$-arrestin sequences, including Ser$^{412}$ for $\beta$-arrestin1 (*Lin et al., 1997*; *Barthet et al., 2009*; *Lin et al., 1999*) and Thr$^{276}$, Ser$^{361}$ and Thr$^{383}$ for $\beta$-arrestin2 (*Lin et al., 2002*; *Paradis et al., 2015*; *Kim et al., 2002*). These phosphorylation events affect GPCR internalization and/or sequestration and, consequently, steady-state level of GPCR cell-surface expression: phosphorylation of $\beta$-arrestin1 at Ser$^{412}$ by Erk1/2 as well as the phosphorylation of $\beta$-arrestin2 at both Ser$^{361}$ and Thr$^{383}$ reduce their ability to induce internalization of $\beta_2$-adrenergic receptor (*Lin et al., 1997*, *1999*, *2002*), whereas phosphorylation of $\beta$-arrestin2 at Ser$^{14}$ and Thr$^{276}$ promotes intracellular sequestration of CXCR4 receptor (*Paradis et al., 2015*). These results indicate that $\beta$-arrestin2 phosphorylation exerts contrasting effects on GPCR trafficking that may depend on the nature of the phosphorylated residue(s) and of the GPCR. Another phosphorylated site (Ser$^{178}$) was identified in rat/mouse $\beta$-arrestin2 and its phosphorylation state affects endosomal trafficking of various GPCRs (*Khoury et al., 2014*). However, this serine residue is not conserved in other species including human, suggesting different regulations of endosomal GPCR trafficking between species. Additional phosphorylated residues have been found on $\beta$-arrestin2 in large-scale phosphoproteomics screens, but their functional relevance remains to be established (*Pighi et al., 2011*; *Sharma et al., 2014*; *Ballif et al., 2008*; *Villén et al., 2007*; *Choudhary et al., 2009*; *Jørgensen et al., 2009*). In contrast to its role in GPCR trafficking, the influence of $\beta$-arrestin phosphorylation on GPCR-operated $\beta$-arrestin-dependent signaling such as Erk1/2 activation remains unexplored.

Here, we investigated the impact on $\beta$-arrestin1 and 2 phosphorylation of expression/stimulation of 5-HT$_{2C}$ and 5-HT$_4$ receptors, two GPCRs that differ in their $\beta$-arrestin dependency to promote Erk1/2 activation, using high-resolution mass spectrometry. We identified several previously described as well as novel phosphorylated residues on $\beta$-arrestin2, while only one phosphorylated site was found on $\beta$-arrestin1. Of these, only phosphorylation of $\beta$-arrestin2 at Thr$^{383}$ exhibited a strong increase upon 5-HT$_{2C}$ receptor stimulation. Furthermore, Thr$^{383}$ was poorly phosphorylated in cells expressing 5-HT$_4$ receptor and its phosphorylation was only slightly enhanced by agonist treatment. These findings prompted functional studies to evaluate the influence of this phosphorylation event in Erk1/2 phosphorylation elicited by stimulation of 5-HT$_{2C}$ receptor and other GPCRs known to engage Erk signaling in a $\beta$-arrestin-dependent manner, in comparison with 5-HT$_4$ receptor stimulation, which induces Erk1/2 activation through a $\beta$-arrestin-independent mechanism.

## Results

### 5-HT$_{2C}$ and 5-HT$_4$ receptor stimulation induce distinct patterns of β-arrestin phosphorylation

To characterize in a global manner the impact of 5-HT$_{2C}$ and 5-HT$_4$ receptor expression/stimulation on the phosphorylation state of $\beta$-arrestins, YFP-tagged versions of $\beta$-arrestin1 or $\beta$-arrestin2 were co-expressed with Myc-tagged 5-HT$_{2C}$ or 5-HT$_4$ receptors in HEK-293 cells. Cells were then treated with vehicle or 5-HT for 5 or 30 min. $\beta$-arrestins were immunoprecipitated using GFP nanobodies coupled to sepharose beads (GFP-Trap), resolved by SDS-PAGE, detected by colloidal Coomassie blue staining (*Figure 1—figure supplement 1*) and digested in-gel with trypsin. Analysis of the resulting peptides by nano-LC-MS/MS yielded 88% and 85% sequence coverage for $\beta$-arrestin1 and $\beta$-arrestin2, respectively, with a p-value threshold of 0.01 for peptide identification (*Figure 1—figure supplement 2*).

Only one phosphorylated site (Thr$^{374}$) was identified on $\beta$-arrestin1, but the corresponding phosphorylated peptide was only detected in cells expressing 5-HT$_4$ receptor, and with a low phosphorylation index, as estimated by dividing the MS signal intensity of the phosphorylated peptide by the sum of MS signal intensities of the phosphorylated and the corresponding non-phosphorylated peptide (*Figure 1—source data 1*, see also the fragmentation spectrum that pinpoints the position of the phosphorylation site on *Figure 1—figure supplement 3*). Moreover, label-free quantification of the MS signal of this phosphorylated peptide in cells treated or not with 5-HT indicated that Thr$^{374}$ phosphorylation is not affected by 5-HT exposure (*Figure 1—source data 1*). Previous studies using a phosphorylated site-specific antibody have identified Ser$^{412}$ as an additional phosphorylated residue on $\beta$-arrestin1 (*Lin et al., 1997*; *Barthet et al., 2009*; *Lin et al., 1999*). They also indicated that Ser$^{412}$ phosphorylation state is modified upon activation of various GPCRs, including the 5-HT$_4$ receptor (*Barthet et al., 2009*). Our MS/MS analyses did not identify this phosphorylated residue, but a low-intensity ion signal corresponding to the theoretical m/z of a peptide comprising the phosphorylated Ser$^{412}$ was detected in cells expressing the 5-HT$_4$ receptor and exposed for 30 min to 5-HT.

Six phosphorylated residues (Thr$^{178}$, Ser$^{194}$, Ser$^{267/268}$, Ser$^{281}$, Ser$^{361}$ and Thr$^{383}$) were identified on $\beta$-arrestin2 (*Figure 1A*, *Figure 1—source data 1* and *Figure 1—figure supplements 3–5*). Note that the fragmentation spectrum of the CPVAQLEQDDQVSPp(S$^{267}$S$^{268}$)TFCK peptide did not provide enough information to discriminate between a phosphorylation at Ser$^{267}$ or at Ser$^{268}$, but the precursor peptide m/z clearly showed that it carries only one phosphorylated residue (*Figure 1—figure supplement 3*). Relative quantification of MS signals of $\beta$-arrestin2 phosphorylated peptides showed that the phosphorylation of these residues except for Thr$^{383}$ was not affected by 5-HT$_{2C}$ or 5HT$_4$ receptor expression or stimulation (*Figure 1—source data 1*). In cells expressing the 5-HT$_{2C}$ receptor, Thr$^{383}$ phosphorylation showed a marked elevation after a 5-min 5-HT stimulation, which persisted 30 min after the onset of the treatment. Thr$^{383}$ phosphorylation slightly but not significantly increased after the 30-min 5-HT challenge in cells expressing 5-HT$_4$ receptor and was much less pronounced than that measured in presence of 5-HT$_{2C}$ receptor (*Figure 1B* and *Figure 1—source data 1*). We next examined whether 5-HT$_{2C}$ receptor stimulation likewise induces phosphorylation of endogenous $\beta$-arrestin2 at Thr$^{383}$, but did not detect any ion signal corresponding to the phosphorylated peptide comprising this residue in cells not transfected with the YFP-$\beta$-arrestin2 plasmid. We thus generated a polyclonal antibody that specifically recognizes $\beta$-arrestin2 phosphorylated at Thr$^{383}$ (see *Figure 2C* and *Figure 2—source data 1*). Western blot experiments using this antibody showed that 5-HT$_{2C}$ receptor stimulation induces a substantial increase in endogenous $\beta$-arrestin2 phosphorylation at Thr$^{383}$, while activation of the 5-HT$_4$ receptor did not affect its phosphorylation level (*Figure 1C* and *Figure 1—source data 2*), corroborating the MS/MS analysis of ectopic $\beta$-arrestin2 phosphorylation.

## Role of MEK in $\beta$-arrestin2 phosphorylation at Thr$^{383}$ elicited by 5-HT$_{2C}$ receptor stimulation

Given the strong enhancement of Thr$^{383}$ phosphorylation induced by the stimulation of 5-HT$_{2C}$ receptor that parallels the $\beta$-arrestin2-dependent receptor-operated activation of Erk1/2 (*Figure 1* and *Figure 1—figure supplement 1*), we paid particular attention to the phosphorylation of this residue, which is located in an unfolded region of the carboxy-terminal region of $\beta$-arrestin2 easily accessible to protein kinases (*Figure 1A*). In an effort to identify kinase(s) involved in its phosphorylation, we first used Group-based Prediction System (GPS, v2.1), an algorithm for kinase consensus search that classifies protein kinases into a hierarchical structure with four levels and trains against the PhosphoELM database, in order to determine individual false discovery rates for each of them (*Xue et al., 2008*). GPS search revealed that Thr$^{383}$ is a strong consensus for phosphorylation by casein kinase 2 (CK2 GPS score 5.3), consistent with previous findings (*Lin et al., 2002*; *Kim et al., 2002*).

However, neither basal nor 5-HT$_{2C}$ receptor-elicited Thr$^{383}$ phosphorylation was affected by treating cells with the selective cell-permeable CK2 pharmacological inhibitor tetrabromocinnamic acid (TBCA, *Figure 2—figure supplement 1* and *Figure 2—figure supplement 1—source data 1*) (*Pagano et al., 2007*), indicating a marginal contribution of CK2 in the phosphorylation of this residue in living HEK-293 cells. GPS search also suggested that Thr$^{383}$ is a potential site for phosphorylation by MAP2K1 (MEK1, GPS score 6.0). Numerous studies have shown that $\beta$-arrestins associated

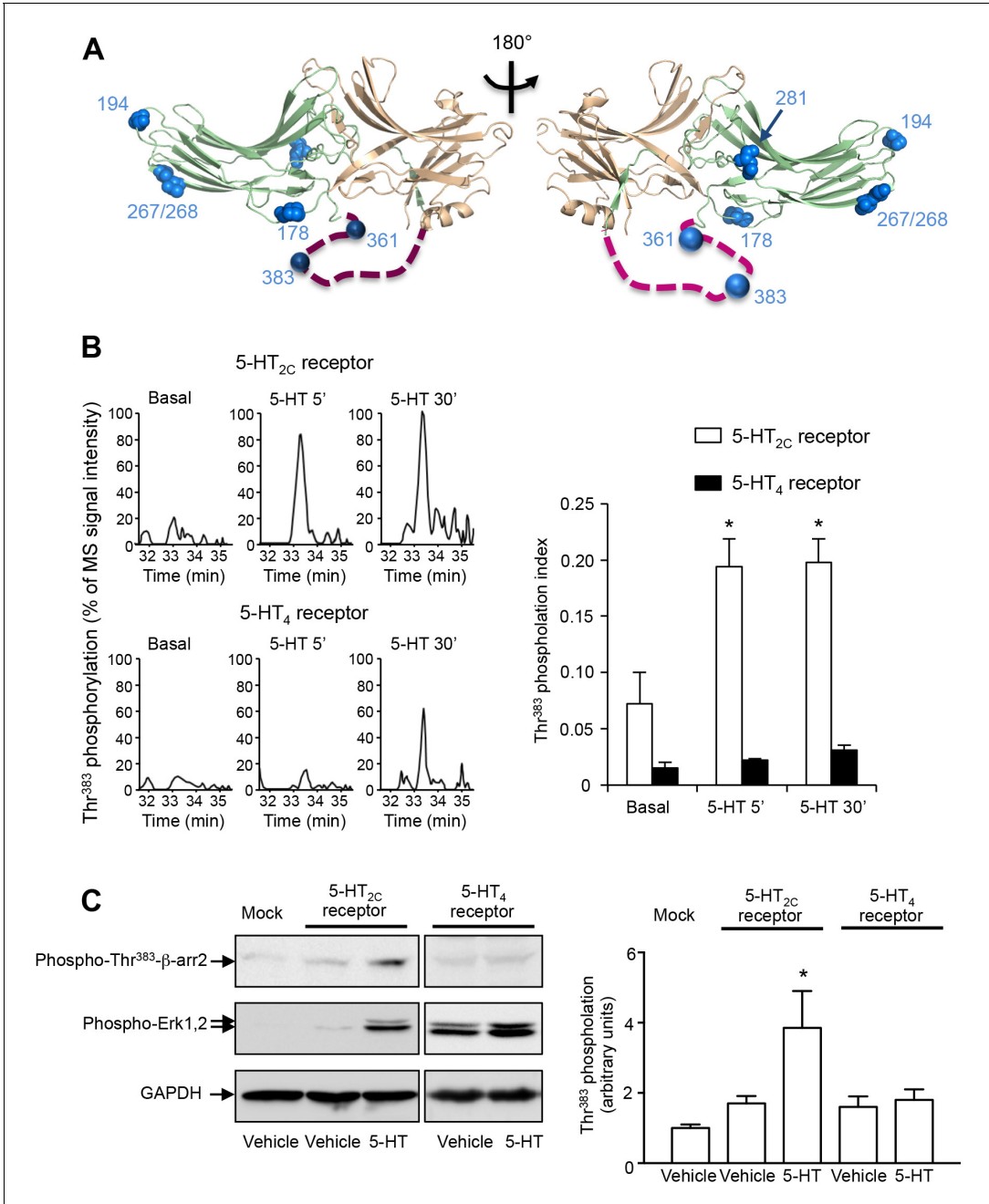

**Figure 1.** 5-HT$_{2C}$ and 5-HT$_4$ receptor stimulation promotes $\beta$-arrestin2 phosphorylation in HEK-293 cells. (**A**) Ribbon diagram of rat $\beta$-arrestin2 showing the position of phosphorylated residues identified by LS-MS/MS. (**B**) Representative extracted ion chromatograms of the EIDIPVDTNLIEFDTNYAp$^{383}$TDDDIVFEDFAR peptide from YFP-tagged $\beta$-arrestin2 in cells expressing 5-HT$_{2C}$ or 5-HT$_4$ receptor and challenged with vehicle (Basal) or 5-HT (1 and 10 μM, respectively) for 5 or 30 min. Two other independent experiments performed on different sets of cultured cells yielded similar results. The histogram represents the means ± SEM of ion signal intensities of the peptide obtained in the three experiments. (**C**) 5-HT$_{2C}$ or 5-HT$_4$ receptor expressing cells were treated as in (**B**). Erk1,2 activation and Thr$^{383}$ phosphorylation were assessed by Western blotting using the anti-phospho-Thr$^{202}$/Tyr$^{204}$-Erk1/2 and the anti-phospho-Thr$^{383}$ $\beta$-arrestin2 antibody, respectively. The histogram shows the means ± SEM of the anti-phospho-Thr$^{383}$ $\beta$-arrestin2 immunoreactive signals (expressed in arbitrary unit) obtained in three independent experiments performed on different sets of cultured cells. One-way ANOVA: (**B**) $F_{(5,12)}=7.544$, $p=0.0020$; (**C**) $F_{(4,10)} = 4.417$, $p=0.0259$. *$p<0.05$ vs. corresponding vehicle.

The following source data and figure supplements are available for figure 1:

**Source data 1.** List of phosphorylated peptides identified from purified $\beta$-arrestin1 and $\beta$-arrestin2 by nano-LC-MS/MS.

*Figure 1 continued on next page*

*Figure 1 continued*

**Source data 2.** This file contains raw values used to build *Figure 1C*.

**Figure supplement 1.** Purification of YFP-$\beta$-arrestin1 and YFP-$\beta$-arrestin2 co-expressed with 5-HT$_{2C}$ or 5-HT$_4$ receptor in HEK-293 cells.

**Figure supplement 2.** Sequence coverage of $\beta$-arrestin1 and $\beta$-arrestin2 obtained by LC-MS/MS.

**Figure supplement 3.** Tandem mass spectra of EVPESETPVDpT$^{374}$NLIELDT NDDDIVFEDFAR, CPVAQLEQDDQVSPp(S$^{267}$S$^{268}$)TFCK and EIDIPVDTNLIEFD TNYApT$^{383}$DDDIVFEDFAR phosphorylated peptides identified from YFP-tagged $\beta$-arrestin1 and $\beta$-arrestin2 transiently co-expressed with 5-HT$_{2C}$ receptor in HEK-293 cells and immunoprecipitated using the GFP Trap kit.

**Figure supplement 4.** Tandem mass spectra of HFLMpS$^{194}$DRR, KVQFAPE pT$^{178}$PGPQPSAETTR and PHDHITLPRPQpS$^{361}$APR phosphorylated peptides identified from YFP-tagged $\beta$-arrestin1 and $\beta$-arrestin2 transiently co-expressed with 5-HT$_{2C}$ receptor in HEK-293 cells and immunoprecipitated using the GFP Trap kit.

**Figure supplement 5.** Tandem mass spectra of VQFAPEpT$^{178}$PGPQPSAET TR, VYTITPLLpS$^{281}$DNR, VYTITPLLpS$^{281}$DNREK phosphorylated peptides identified from YFP-tagged $\beta$-arrestin1 and $\beta$-arrestin2 transiently co-expressed with 5-HT$_{2C}$ receptor in HEK-293 cells and immunoprecipitated using the GFP Trap kit.

with a GPCR can interact with different signaling proteins including c-Src and several proteins of the Erk signaling module (Raf/MEK/Erk), and that these interactions depend on $\beta$-arrestin conformational state (*Xiao et al., 2004*; *Gurevich and Gurevich, 2003*).

In line with these findings, we have recently modeled the complex between a GPCR, $\beta$-arrestin2, c-Src and the Erk module from the 3D structure of each partner, using the Protein-Protein cOmplexes 3D structure pRediction (PRIOR) docking algorithm, and validated experimentally the previously unknown interaction regions (*Bourquard et al., 2015*). This model predicts that the unfolded C-terminal part of $\beta$-arrestin comprising Thr$^{383}$ is located in the vicinity of the MEK active site (*Figure 2A*), indicating a possible role of MEK bound to $\beta$-arrestin2 in its phosphorylation. Based on this model, we hypothesized that the mechanism of $\beta$-arrestin-dependent activation of Erk consists in the following steps: (i) Raf and MEK assemble to $\beta$-arrestin2 bound to agonist-stimulated receptor, resulting in MEK activation, (ii) MEK phosphorylates $\beta$-arrestin2 at Thr$^{383}$, resulting in a large conformational change of the $\beta$-arrestin2 region comprising residues 350–393, (iii) the receptor C-terminus associates with $\beta$-arrestin2 using the region previously occupied by the 383–393 $\beta$-strand within the $\beta$-arrestin2 C-terminal tail and (iv) Erk binds to the complex and can be activated by MEK (*Figure 2A*). Consistent with this hypothesis, pretreatment of cells with the MEK pharmacological inhibitor U0126 (5 µM) abolished Thr$^{383}$ phosphorylation elicited by 5-HT$_{2C}$ receptor stimulation (*Figure 2B* and *Figure 2—source data 1*). Expression of a MEK1 dominant-negative inhibitor (MEK1DN) also strongly reduced Thr$^{383}$ phosphorylation state (*Figure 2B*, see also *Figure 2—figure supplement 2* for controls showing strong MEK inhibition induced by both approaches, as assessed by monitoring phosphorylation of Erk1/2). In contrast, pretreating cells with FR180204 (10 µM), a selective pharmacological inhibitor of Erk1/2, did not affect Thr$^{383}$ phosphorylation (*Figure 2B*). The ability of FR180204 treatment to inhibit Erk1/2 activity without affecting MEK activity in our experimental conditions was assessed by monitoring phosphorylation of Erk1/2 and their substrate Elk1 (*Figure 2—figure supplement 2*). Collectively, these results identify MEK as the major kinase involved in phosphorylation of $\beta$-arrestin2 at Thr$^{383}$ upon 5-HT$_{2C}$ receptor stimulation. To further confirm the ability of MEK to directly phosphorylate this residue, we performed an in vitro kinase assay using purified active MEK1 and YFP-$\beta$-arrestin2 purified from transfected HEK-293 cells as substrate, followed by Western blotting using the anti-phospho-Thr$^{383}\beta$-arrestin2 antibody. MEK1 induced a strong elevation in the immunoreactive signal that was abolished by adding U0126 in the incubation medium (*Figure 2C*). As expected, no increase in immunoreactive signal was observed in experiments using YFP-Thr$^{383}$Ala $\beta$-arrestin2 mutant as substrate, thus validating the specificity of the antibody for $\beta$-arrestin2 phosphorylated at Thr$^{383}$. This result was confirmed using a radioactive kinase assay, which also showed comparable efficacy of MEK1 to promote phosphorylation of

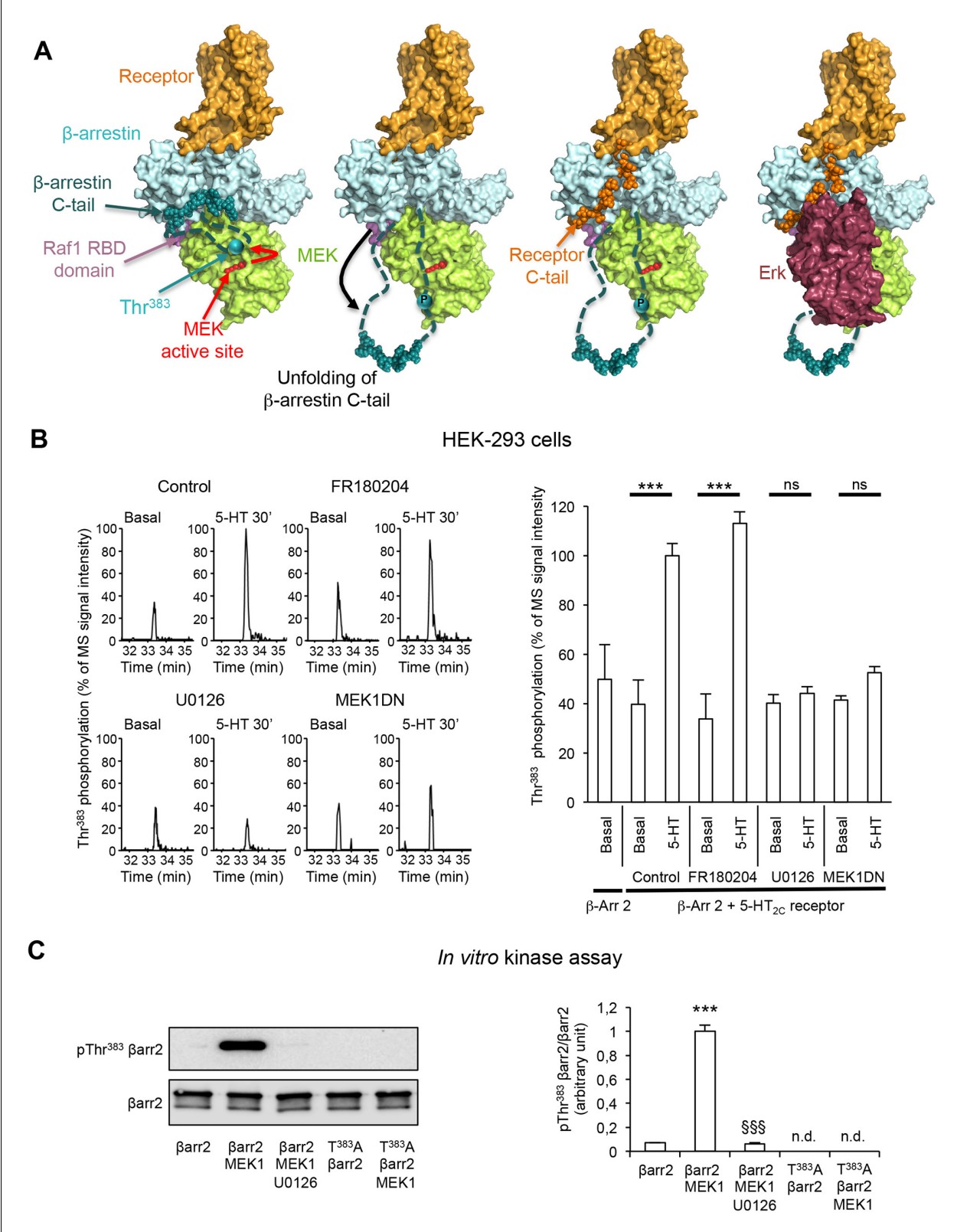

**Figure 2.** Role of MEK in the phosphorylation of β-arrestin2 at Thr383 elicited by 5-HT2C receptor stimulation. (**A**) Mechanistic model of assembly of the 5-HT2C receptor/β-arrestin2/Erk module. Color code: receptor in orange, MEK in green, β-arrestin2 core in pale cyan and C-tail in cyan (the regions 351–384 and 394–419, which are not visible in 3D structure are represented by dashed lines, the region 385–393 is represented by spheres), Erk in dark red, Raf-1 RBD domain in pink. In this model, we hypothesize that Thr383 phosphorylation by MEK takes place within the assembled receptor/β-arrestin/

*Figure 2 continued*

Raf/MEK complex and results in a movement of β-arrestin2 unfolded 350–393 segment away from the first β-strand of β-arrestin, leaving space for further interaction with the receptor C-terminal domain (orange spheres) and recruitment of Erk, and its subsequent phosphorylation by MEK. For the clarity of the figure, the extremity of the β-arrestin C-tail is represented by spheres even in its unfolded state, although the real 3D structure is unknown. (B) Representative extracted ion chromatograms of the peptide in cells expressing 5-HT$_{2C}$ receptor, pretreated with either vehicle (control) or FR180204 (10 μM for 18 hr) or U0126 (5 μM for 30 min) or coexpressing MEK1 dominant-negative mutant (MEK1DN), and challenged with vehicle (Basal) or 5-HT (1 μM) for 30 min. The histogram represents the means ± SEM of the corresponding ion signal intensities (normalized to values in 5-HT-stimulated cells in Control condition) obtained in three independent experiments. One-way ANOVA: $F_{(8,18)} = 15.69$, $p<0.0001$. ***$p<0.001$ *vs.* corresponding basal value. (C) YFP-tagged β-arrestin2 (wild-type or Thr$^{383}$Ala mutant) purified from transfected HEK-293 cells was incubated with active MEK1 for 15 min at 37°C. When indicated, U0126 (5 μM) was included in the incubation medium. Thr$^{383}$ phosphorylation was assessed by sequential immunoblotting with the antibody raised against phospho-Thr$^{383}$ β-arrestin2 and the anti-β-arrestin2 antibody. Means ± SEM of results from four independent experiments are shown on the histogram. n.d.: not detectable. One-way ANOVA: $F_{(2,9)} = 352.2$, $p<0.0001$. ***$p<0.001$ *vs.* immunoreactive signal in absence of MEK; §§§ $p<0.001$ *vs.* corresponding condition in absence of U0126.

The following source data and figure supplements are available for figure 2:

**Source data 1.** This file contains raw values used to build *Figure 2B*, *C*.
**Figure supplement 1.** Thr$^{383}$ phosphorylation is not mediated by casein kinase 2.
**Figure supplement 1—source data 1.** This file contains raw values used to build *Figure 2—figure supplement 1*.
**Figure supplement 2.** Impact of MEK and Erk1/2 pharmacological inhibitors and of co-expression of MEK dominant-negative mutant on 5-HT$_{2C}$ receptor-operated Erk1/2 and Elk1 phosphorylation.
**Figure supplement 3.** In vitro phosphorylation of β-arrestin2 versus Erk2 by MEK1.
**Figure supplement 3—source data 1.** This file contains raw values used to build *Figure 2—figure supplement 3*.

purified β-arrestin2 and its canonical substrate Erk2 in vitro (*Figure 2—figure supplement 3* and *Figure 2—figure supplement 3—source data 1*).

## β-arrestin2 phosphorylation at Thr$^{383}$ neither affects translocation of β-arrestin2 to 5-HT$_{2C}$ and 5-HT$_4$ receptors nor agonist-induced receptor internalization

To explore whether Thr$^{383}$ phosphorylation affects the recruitment of β-arrestin2 by 5-HT$_{2C}$ and 5-HT$_4$ receptors, we monitored interaction between each receptor and either wild type or Thr$^{383}$Ala or Thr$^{383}$Asp β-arrestin2 mutants using a BRET-based assay. Exposure of cells to 5-HT induced a strong increase in the BRET signal between 5-HT$_{2C}$-YFP or 5-HT$_4$-YFP receptor and Rluc-β-arrestin2 (*Figure 3A and B* and *Figure 3—source data 1*). The 5-HT-elicited BRET signal was similar in cells expressing wild type β-arrestin2, Thr$^{383}$Ala or Thr$^{383}$Asp β-arrestin2 mutants, indicating that Thr$^{383}$ phosphorylation does not affect β-arrestin2 translocation to either receptor. Likewise, 5-HT treatment decreased cell surface expression of 5-HT$_{2C}$ and 5-HT$_4$ receptors to a similar extent in cells expressing wild type β-arrestin2, Thr$^{383}$Ala or Thr$^{383}$Asp β-arrestin2 mutants (*Figure 3C and D*), indicating that Thr$^{383}$ phosphorylation does not affect agonist-induced internalization of these receptors.

## Impact of Thr$^{383}$ phosphorylation upon β-arrestin2 conformational changes elicited by 5-HT$_{2C}$ and 5-HT$_4$ receptor stimulation

β-arrestins are known to undergo important conformational rearrangements upon translocation to agonist-stimulated GPCRs that are essential for their downstream action (*Xiao et al., 2004*; *Gurevich and Gurevich, 2014*). To explore the impact of Thr$^{383}$ phosphorylation on conformational changes of β-arrestin2 induced by 5-HT$_{2C}$ receptor activation *vs.* 5-HT$_4$ receptor activation, we used an optimized intramolecular BRET biosensor consisting of β-arrestin2 sandwiched between the

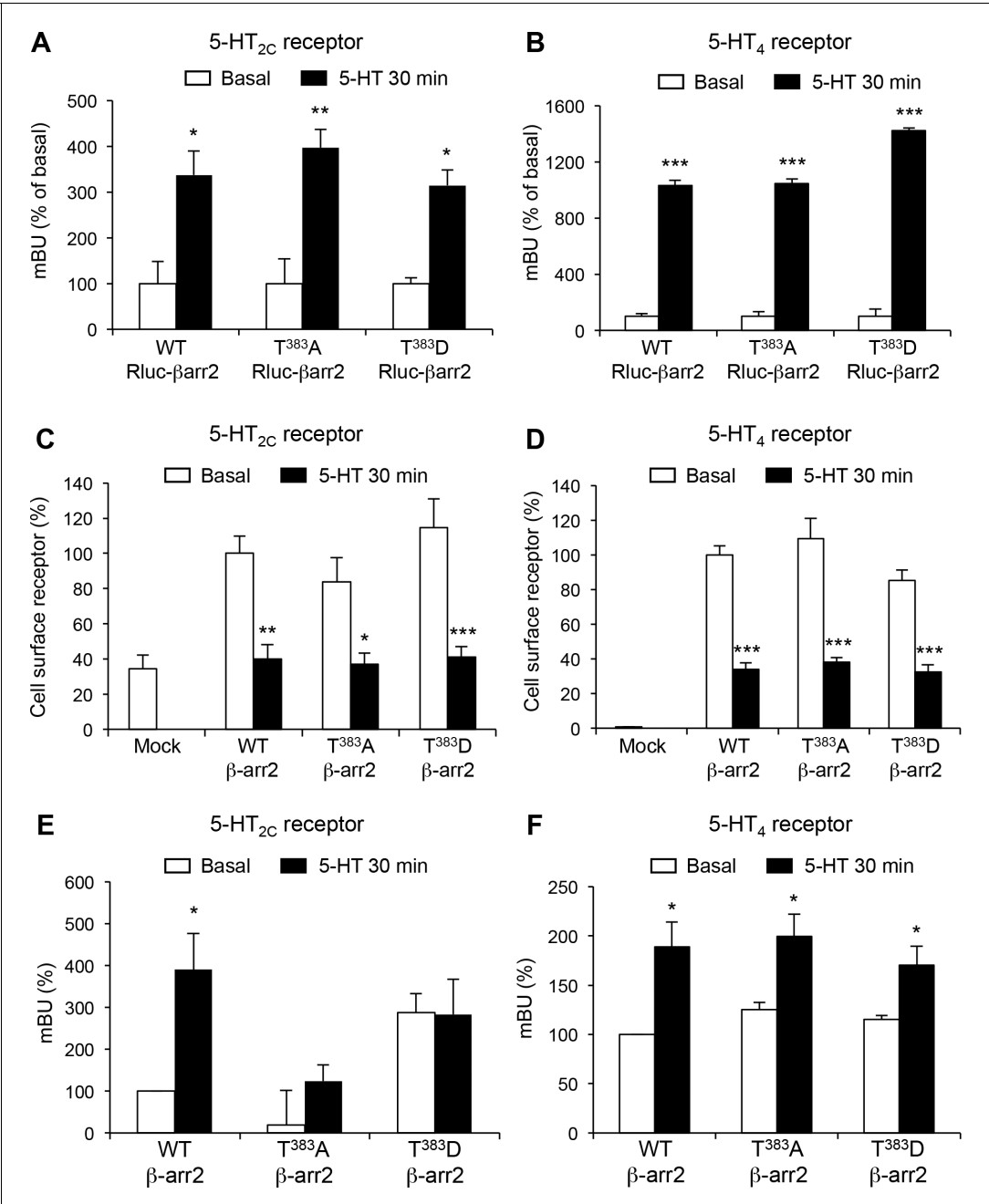

**Figure 3.** Thr[383] phosphorylation underlies $\beta$-arrestin2 conformational rearrangement elicited by 5-HT$_{2C}$ receptor stimulation. (A, B) Translocation of wild type (WT), T[383]A and T[383]D Rluc-$\beta$-arrestin2 to Myc-5-HT$_{2C}$-YFP (A) or Myc-5-HT$_4$-YFP (B) receptors in cells treated with either vehicle (Basal) or 1 or 10 $\mu$M 5-HT, respectively, was measured by BRET. Data represent the mean ± SEM of values obtained in three independent experiments and were normalized to the BRET signals measured in 5-HT-stimulated cells expressing WT Rluc $\beta$-arrestin2. (C, D) Cell surface expression of receptors was measured in the same experimental condition by ELISA using anti-Myc antibody. Data are the mean ± SEM of values obtained in three independent experiments. They were normalized to total receptor expression level and are expressed in % of basal receptor level at the cell surface in cells expressing WT $\beta$-arrestin2. (E, F) Conformational arrangement of WT, T[383]A and T[383]D double brilliance Rluc8-$\beta$-arrestin2-RGFP elicited by 5-HT$_{2C}$ and 5-HT$_4$ receptor stimulation by 5-HT (1 and 10 $\mu$M, respectively). Equivalent expression of each BRET sensor was verified by ELISA. Data represent the mean ± SEM of values obtained in three independent experiments and were normalized to the basal intra-molecular BRET signal in cells expressing WT Rluc8-$\beta$-arrestin2-RGFP. One-way ANOVA: A, $F_{(5,12)}=10.75$, p=0.0004; B, $F_{(5,12)}=320.9$, p<0.001; C, $F_{(6,14)}=10.82$, p<0.0001; D, $F_{(6,14)}=48.52$, p<0.0001; E, $F_{(5,12)}=5.136$, p=0.0095; F, $F_{(5,12)}=6.436$, p=0.004. *p<0.05, **p<0.01 ***p<0.001 *vs.* corresponding basal.

The following source data is available for figure 3:

*Figure 3 continued on next page*

*Figure 3 continued*

**Source data 1.** This file contains raw values used to build *Figure 3*.

*Renilla* green fluorescent protein (RGFP) and the *Renilla* luciferase variant Rluc8 (*Kamal et al., 2009*; *Charest et al., 2005*). When β-arrestin2 conformation changes, the distance between Rluc8 and RGFP increases or decreases leading to a BRET signal decrease or increase. Treating cells coexpressing 5-HT$_{2C}$ receptor and WT RLuc8-β-arrestin2-RGFP with 5-HT increased the BRET signal. No BRET increase was detected upon agonist exposure in cells coexpressing the receptor and T$^{383}$A RLuc8-β-arrestin2-RGFP (*Figure 3E*). Mutation of Thr$^{383}$ into aspartate in RLuc8-β-arrestin2-RGFP also resulted in an increase in BRET signal to a level similar to the one measured in cells expressing the wild type probe and exposed to 5-HT, and 5-HT treatment did not further enhance this elevated BRET signal (*Figure 3E*). In contrast, substitution of Thr$^{383}$ by alanine or aspartate did not affect the increase in RLuc8-β-arrestin2-RGFP BRET signal induced by 5-HT treatment in cells expressing 5-HT$_4$ receptors (*Figure 3F*). Collectively, these results indicate that β-arrestin2 conformational change elicited by 5-HT$_{2C}$ receptor stimulation, but not 5-HT$_4$ receptor stimulation, depends on Thr$^{383}$ phosphorylation.

## β-arrestin2 phosphorylation at Thr$^{383}$ is essential for Erk1/2 translocation to the 5-HT$_{2C}$ receptor/β-arrestin2 complex

As previously hypothesized, it can be envisioned that the unfolded C-terminal region of β-arrestin2 occupies the docking site of Erk1/2 located in the vicinity of MEK catalytic site in the receptor/β-arrestin2 complex (*Figure 2A*). Consequently, Thr$^{383}$ phosphorylation status might affect Erk1/2 recruitment. In fact, our model predicts that Thr$^{383}$ phosphorylation should induce a movement of the β-arrestin2 C-terminal region away from the complex leaving space for further recruitment of Erk (*Figure 2A*). To explore this possibility, we compared the ability of wild type and Thr$^{383}$Ala and Thr$^{383}$Asp β-arrestin2 to recruit Erk1/2 upon 5-HT$_{2C}$ receptor stimulation by co-immunoprecipitation. Treating 5-HT$_{2C}$ receptor-expressing cells with 5-HT increased the amount of Erk1/2 co-immunoprecipitated with β-arrestin2, an effect abolished by mutating Thr$^{383}$ into Ala (*Figure 4A* and *Figure 4—source data 1*). In contrast, mutation of Thr$^{383}$ into aspartate increased Erk1/2 recruitment by β-arrestin2 even in the absence of the agonist and receptor stimulation did not induce any additional effect (*Figure 4A*). Further supporting a role of Thr$^{383}$ phosphorylation by MEK, treatment of cells with U0126 prevented recruitment of Erk1,2 by β-arrestin2 induced by 5-HT$_{2C}$ receptor stimulation (*Figure 4—figure supplement 1* and *Figure 4—figure supplement 1—source data 1*). 5-HT did not promote Erk1/2 recruitment by β-arrestin2 in cells coexpressing 5-HT$_4$ receptor and either wild type or mutated forms of β-arrestin2 (*Figure 4B*). Collectively, these results suggest that Thr$^{383}$ phosphorylation by MEK promotes a conformational change of β-arrestin2 that facilitates Erk recruitment by the 5-HT$_{2C}$ receptor/β-arrestin2 complex, whereas it has no influence upon its translocation to the 5-HT$_4$ receptor/β-arrestin2 complex.

## Role of Thr$^{383}$ phosphorylation in 5-HT$_{2C}$ receptor-operated Erk1/2 signaling

In line with the role of Thr$^{383}$ phosphorylation in Erk recruitment by the 5-HT$_{2C}$ receptor/β-arrestin2 complex, we next explored the influence of this phosphorylation event in receptor-operated Erk1/2 activation, which has previously been reported to depend on β-arrestin2 (*Labasque et al., 2008*). The ability of 5-HT (both 5- and 30 min exposures) to promote Erk1/2 phosphorylation was strongly reduced in cells co-expressing 5-HT$_{2C}$ receptor and T$^{383}$A β-arrestin2, compared with cells expressing WT β-arrestin2 (*Figure 4C*). Likewise, Erk1/2 phosphorylation was reduced in non-stimulated cells expressing T$^{383}$A β-arrestin2, while expression of T$^{383}$D β-arrestin2 increased basal Erk1/2 phosphorylation to a level that was not further increased by agonist treatment. Altogether, these results identify Thr$^{383}$ phosphorylation as a key event contributing to basal and 5-HT$_{2C}$ receptor-operated Erk1/2 activation. In contrast and consistent with the β-arrestin-independent 5-HT$_4$ receptor-operated Erk signaling (*Barthet et al., 2007*), mutating Thr$^{383}$ into alanine or aspartate did not affect Erk1/2 phosphorylation elicited by 5-HT$_4$ receptor stimulation (*Figure 4D*).

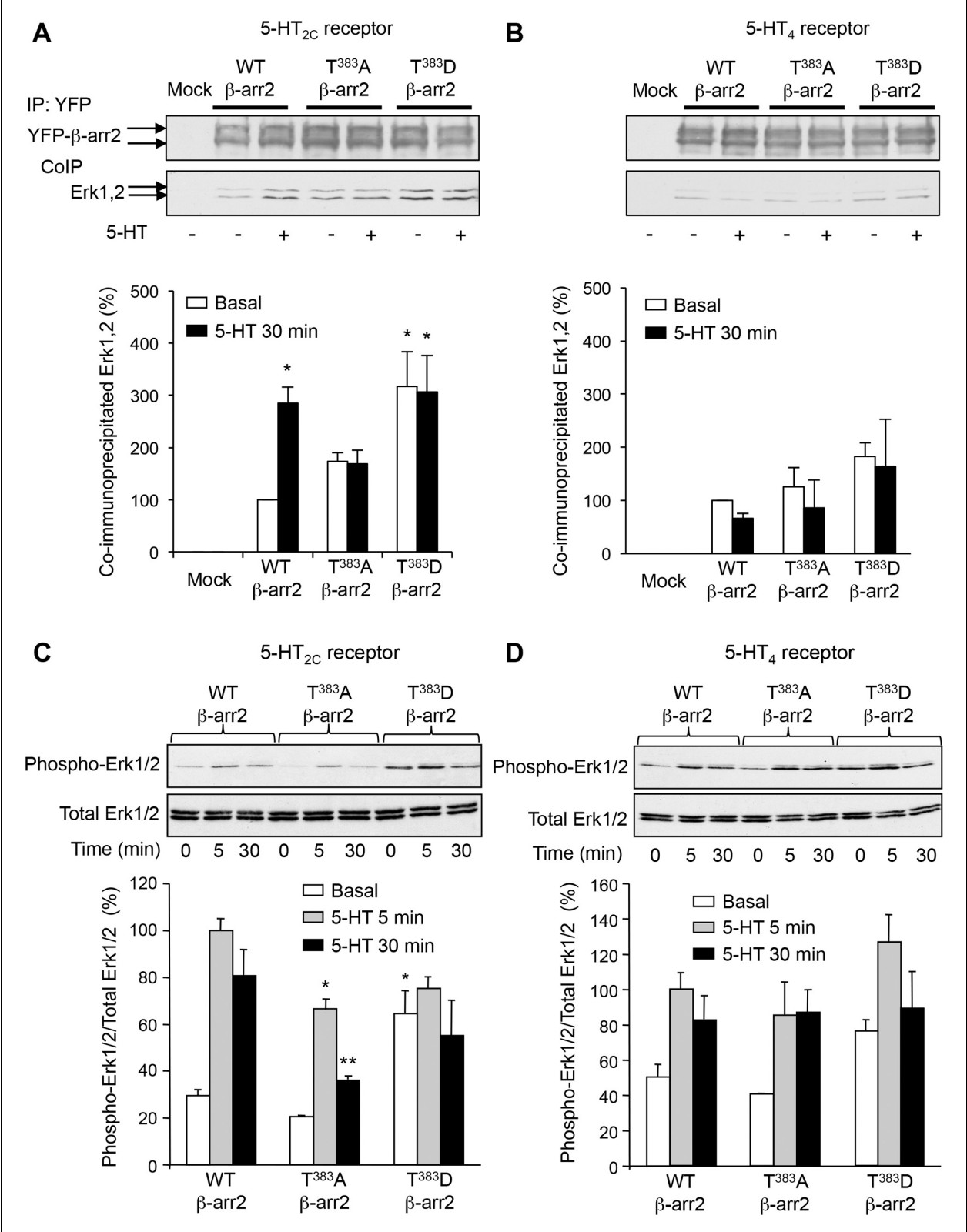

**Figure 4.** Phosphorylation of β-arrestin2 at Thr[383] is a necessary step in Erk1/2 recruitment to β-arrestin2 and engagement of Erk signaling by 5-HT$_{2C}$ receptor. (A, B) Recruitment of Erk1/2 to WT, T[383]A and T[383]D YFP-β-arrestin2 in cells expressing 5-HT$_{2C}$ or 5-HT$_4$ receptor and exposed or not to 5-HT (1 and 10 µM, respectively) was assessed by co-immunoprecipitation. Immunoblots representative of three independent experiments are illustrated. The histograms represent the means ± SEM of Erk1/2 immunoreactive signals in immunoprecipitates, assessed by densitometric analysis, obtained in the

*Figure 4 continued on next page*

*Figure 4 continued*

three experiments. They were normalized to the amount of YFP-$\beta$-arrestin2 immunoprecipitates and expressed in % of basal level measured in cells expressing WT $\beta$-arrestin2. *p<0.05 vs. basal value in cells expressing WT $\beta$-arrestin2. (C, D) Erk1/2 activation in cells co-expressing 5-HT$_{2C}$ or 5-HT$_4$ receptor and either WT, or T$^{383}$A and or T$^{383}$D YFP-$\beta$-arrestin2 and exposed or not to 5-HT (1 and 10 $\mu$M, respectively) for 5 or 30 min was assessed by sequential immunoblotting with the antibody recognizing phospho-Thr$^{202}$/Tyr$^{204}$-Erk1/2 and total Erk1/2. Immunoblots representative of three independent experiments are illustrated. The histograms represent the means ± SEM of values (normalized to the level of phosphorylated Erk1/2 in cells expressing WT $\beta$-arrestin2 and exposed to 5-HT for 5 min) obtained in the three experiments. One-way ANOVA: A, $F_{(5,12)}$=4.305, p=0.00178; B, $F_{(5,12)}$=0.977, p=0.47; C, $F_{(8,18)}$=11.78, p<0001; D, $F_{(8,18)}$=4.998, p=0.0022. *p<0.05, **p<0.01 vs. corresponding value in cells expressing WT $\beta$-arrestin2.

The following source data and figure supplements are available for figure 4:

**Source data 1.** This file contains raw values used to build *Figure 4*.

**Figure supplement 1.** Role of MEK in Erk recruitment to $\beta$-arrestin2.

**Figure supplement 1—source data 1.** This file contains raw values used to build *Figure 4—figure supplement 1*.

## Role of Thr$^{383}$ phosphorylation in Erk1/2 signaling elicited by other GPCRs

We next explored the impact of Thr$^{383}$ phosphorylation on Erk1/2 activation induced by stimulation of other GPCRs known to engage the Erk pathway via the classic dual mechanism involving both G protein- and $\beta$-arrestin-dependent pathways, namely follicle-stimulating hormone (FSH) receptor, $\beta$-adrenergic receptors and chemokine CXCR4 receptor (*Shenoy et al., 2006*; *Kara et al., 2006*; *Sun et al., 2002*). Consistent with our previous findings (*Kara et al., 2006*), silencing $\beta$-arrestin2 expression strongly reduced the level of Erk1/2 phosphorylation elicited by exposing cells transiently expressing FSH receptor to FSH for 30 min, whereas it only slightly but not significantly affected Erk1/2 phosphorylation induced by a 5-min FSH treatment (*Figure 5A*, see also *Figure 5—figure supplement 1*, *Figure 5—source data 1* and *Figure 5—figure supplement 1—Source data 1*). Erk1/2 phosphorylation level measured after a 30 min FSH treatment was also significantly diminished in cells expressing T$^{383}$A $\beta$-arrestin2, compared with WT $\beta$-arrestin2 (*Figure 5D*). Likewise, Erk1/2 phosphorylation elicited by stimulating endogenously expressed $\beta$-adrenergic receptor by isoproterenol or CXCR4 receptor by CXCL12 was reduced to a similar extend by silencing $\beta$-arrestin2 expression (*Figure 5B,C*) or by substituting Thr$^{383}$ by alanine (*Figure 5E,F*). Collectively, these results indicate that Thr$^{383}$ substitution by alanine reproduces the inhibitory effect of $\beta$-arrestin2 down-regulation upon activation of Erk1/2 by the three GPCRs examined, suggesting that phosphorylation of this residue is a general mechanism underlying $\beta$-arrestin-dependent Erk1/2 activation by GPCRs.

## Discussion

In this study, using high-resolution mass spectrometry combined with label-free quantification, we provide for the first time a comprehensive characterization of phosphorylated sites on $\beta$-arrestins 1 and 2 in living cells. We also explored the influence on their phosphorylation status of 5-HT$_{2C}$ and 5-HT$_4$ receptors, two model GPCRs that engage the Erk1/2 pathway through $\beta$-arrestin-dependent and independent mechanisms, respectively. We identified one phosphorylated site on $\beta$-arrestin1 and six on $\beta$-arrestin2, including three previously identified residues (Ser$^{178}$, Ser$^{361}$ and Thr$^{383}$) and three new sites (Ser$^{194}$, Ser$^{267/268}$ and Ser$^{281}$). Two of the $\beta$-arrestin2 phosphorylated sites identified (Ser$^{194}$, Ser$^{267/268}$) are conserved in $\beta$-arrestin1, but we did not detect the corresponding phosphorylated peptides in $\beta$-arrestin1. Although MS/MS does not allow exhaustive identification of peptides present in complex samples, it is unlikely that our analyses, which covered about of 85% of the sequence of both arrestins, missed these phosphorylated residues on $\beta$-arrestin1 or any other robustly phosphorylated residues on both arrestins. Therefore, our results corroborate previous

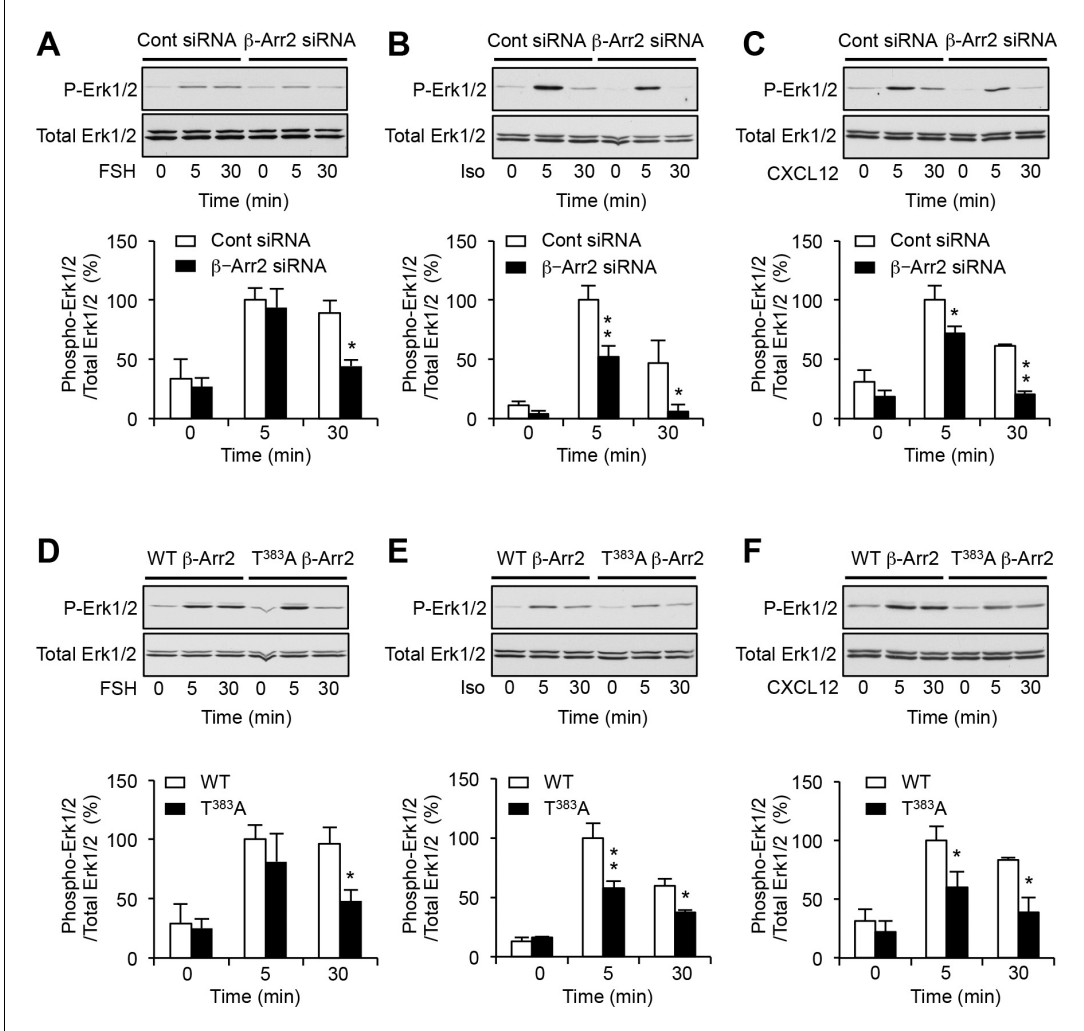

**Figure 5.** Phosphorylation of *β*-arrestin2 at Thr[383] underlies *β*-arrestin-dependent engagement of Erk1/2 signaling by FSH, *β*-adrenergic and CXCR4 receptors. Erk1/2 activation elicited by stimulation for the indicated times of transiently expressed FSH receptor (FSH, 3.3 nM), native *β*-adrenergic (isoproterenol 1 μM) and CXCR4 receptors (CXCL12, 10 nM) in cells transfected with control siRNA or *β*-arrestin2 siRNA (**A–C**) and in cells coexpressing WT or T[383]A *β*-arrestin2 (**D–F**). Erk1/2 activation was assessed by sequential immunoblotting with the antibody recognizing phospho-Thr[202]/Tyr[204]-Erk1/ 2 and total Erk1/2. Immunoblots representative of three independent experiments are illustrated. The histograms represent the means ± SEM of values (normalized to the level of phosphorylated Erk1/2 in cells exposed to agonist for 5 min) obtained in the three experiments. One-way ANOVA: **A**, $F_{(5,12)}$ =8.178, p=0.0014; **B**, $F_{(5,12)}$=16.97, p<0.001; **C**, $F_{(5,12)}$=20.22, p=<0.001; **D**, $F_{(5,12)}$=7.710, p=0.0019; **E**, $F_{(5,12)}$=29.76, p<0.0001; $F_{(5,12)}$=8.695, p=0.0.0012. *p<0.05 and **p<0.01 *vs.* corresponding value in control siRNA transfected cells or cells expressing WT *β*-arrestin2.

The following source data and figure supplements are available for figure 5:

**Source data 1.** This file contains raw values used to build *Figure 5*.

**Figure supplement 1.** Down-regulation of *β*-arrestin2 protein expression in HEK-293 cells using siRNA.

**Figure supplement 1—source data 1.** This file contains raw values used to build *Figure 5—figure supplement 1*.

findings which identified much more phosphorylated residues on $\beta$-arrestin2 than on $\beta$-arrestin1 and suggest that $\beta$-arrestin2 is phosphorylated at higher stoichiometry, compared with $\beta$-arrestin1. Our analyses did not detect, even in 5-HT-treated cells, another site conserved between both $\beta$-arrestins (Ser$^{276}$) that was previously reported as being phosphorylated by recombinant active Erk1 in an in vitro phosphorylation assay followed by MS/MS analysis (*Paradis et al., 2015*). This suggests that in cells expressing 5-HT$_{2C}$ or 5-HT$_4$ receptor, the level of Erk1/2 activity and/or the molecular environment of this serine residue do not allow its phosphorylation at high stoichiometry.

Amongst $\beta$-arrestin2 phosphorylated residues identified in the present study, only the phosphorylation of Thr$^{383}$ was modulated upon agonist stimulation of 5-HT$_{2C}$ receptor and, to a much lesser extent, 5-HT$_4$ receptor. In silico search of kinase-specific phosphorylation sites in $\beta$-arrestin2 sequence identified Thr$^{383}$ as a strong consensus for phosphorylation by CK2, consistent with previous findings which showed that Thr$^{383}$ is phosphorylated by CK2 in vitro (*Kim et al., 2002*), and by MEK. Our results indicate that MEK rather than CK2 is involved in Thr$^{383}$ phosphorylation elicited by 5-HT$_{2C}$ receptor stimulation in living HEK-293 cells, consistent with our docking model of the receptor/$\beta$-arrestin2/MEK complex, which predicts that the unfolded C-terminal part of $\beta$-arrestin comprising Thr$^{383}$ is located in the vicinity of MEK active site.

In an effort to characterize the functional impact of Thr$^{383}$ phosphorylation, we found that its replacement by alanine or aspartate to inhibit or mimic its phosphorylation, did not modify $\beta$-arrestin2 translocation to the receptor nor agonist-induced receptor internalization. This result corroborates previous observations indicating that the sole substitution of Thr$^{383}$ to aspartate or alanine does not affect isoproterenol-induced $\beta_2$-adrenergic receptor internalization (*Lin et al., 2002*). In fact, only the double substitution of Ser$^{361}$ and Thr$^{383}$ to aspartate inhibited agonist-induced receptor internalization, an effect reflecting the reduced ability of this double $\beta$-arrestin mutant to bind to clathrin (*Lin et al., 2002*). It is likely that the phosphorylation stoichiometry of Ser$^{361}$ and Thr$^{383}$ in our experimental conditions, including cells exposed to 5-HT, does not allow such a regulation. Moreover, previous studies showed that $\beta_2$-adrenergic receptor stimulation decreases rather than enhances the phosphorylation of these residues (*Lin et al., 2002*; *Kim et al., 2002*). Accordingly, these studies and our results suggest that $\beta$-arrestin2 phosphorylation at Thr$^{383}$ might only have a minor influence on GPCR trafficking. In contrast, expressing Thr$^{276}$Ala or Ser$^{14}$Ala $\beta$-arrestin2 mutants in $\beta$-arrestin1/2 knockout cells fails to restore agonist-induced CXCR4 receptor sequestration while expressing a $\beta$-arrestin2 mutant harboring an aspartate at these positions is sufficient to decrease cell surface localization in the absence of agonist (*Paradis et al., 2015*). This suggests that Thr$^{276}$ and/or Ser$^{14}$ phosphorylation rather than Thr$^{383}$ phosphorylation might be critical for agonist-induced intracellular sequestration of some GPCRs.

Our results establish a causal link between Thr$^{383}$ phosphorylation and $\beta$-arrestin2 conformational rearrangements, assessed by double brilliance BRET assay, elicited upon 5-HT$_{2C}$ receptor stimulation. They corroborate previous observations suggesting that conformational changes of $\beta$-arrestin do not directly result from its binding to phosphorylated GPCRs, but from downstream events, such as subsequent recruitment of $\beta$-arrestin interacting proteins (*Charest et al., 2005*; *Xiao et al., 2007*). Nonetheless, Thr$^{383}$ phosphorylation dependency of $\beta$-arrestin2 conformational changes seems to also depend on the nature of the GPCR it associates with, as the increase in intramolecular BRET signal measured upon 5-HT$_4$ receptor stimulation was not affected by substituting Thr$^{383}$ by alanine or by aspartate.

The present study also identified Thr$^{383}$ phosphorylation as an essential molecular step allowing Erk1/2 translocation to the 5-HT$_{2C}$/$\beta$-arrestin2/MEK complex and subsequent activation of Erk1/2 signaling upon 5-HT$_{2C}$ receptor stimulation. Based on the current experimental results and our previously validated docking model (*Bourquard et al., 2015*), we propose a molecular mechanism which might explain the role of Thr$^{383}$ phosphorylation in recruitment and activation of Erk at the activated $\beta$-arrestin. In the crystal structures of both $\beta$-arrestin1 (PDB 1G4R) (*Han et al., 2001*) and $\beta$-arrestin2 (PDB 3P2D) (*Zhan et al., 2011*), the segment of the C-terminal tail comprising residues 385–393 folds as a $\beta$-strand, which associates with the first N-terminal $\beta$-strand. The crystal structure of $\beta$-arrestin1 in complex with a C-terminal peptide of the V2 receptor (PDB 4JQI) (*Shukla et al., 2013*), also shows that the C-terminal peptide of the receptor occupies the same location as the 385–393 segment of the inactive $\beta$-arrestin. Consequently, a drastic conformational change of the 385–393 $\beta$-arrestin segment is necessary for the correct association of $\beta$-arrestin with the receptor. Furthermore, our previously reported docking model shows that the C-terminal region of the receptor is

sandwiched between β-arrestin and Erk, and thus has to associate with β-arrestin before its association with Erk (*Bourquard et al., 2015*). Finally, although Thr[383] is not present in the different structures of β-arrestins, the position of Asp[385] shows that Thr[383] is located in the vicinity of β-arrestin-bound MEK's active site, where it can be phosphorylated. In our model (*Figures 2* and *6*), Thr[383] phosphorylation by MEK takes place within the ligand-dependent receptor/β-arrestin/Raf/MEK complex. We propose that this phosphorylation triggers the conformational change of the 350–393 region, allowing the interaction of β-arrestin with the receptor C-terminal domain, the subsequent binding of Erk to the complex and its activation by MEK.

This mechanism, initially established for the 5-HT$_{2C}$ receptor, a GPCR which elicits Erk1/2 signaling mainly through a β-arrestin-dependent mechanism, appears to be conserved among GPCRs that induce Erk1/2 activation through the classic dual mechanism involving both G proteins and β-arrestins, such as β-adrenergic, CXCR4 and FSH receptors. Actually, the sustained Erk1/2 phosphorylation induced by agonist stimulation of these receptors was strongly reduced in cells expressing the Thr[383]Ala β-arrestin2 mutant. In contrast, substitution of Thr[383] by aspartate increased the basal level of Erk1/2 phosphorylation, suggesting that phosphorylated β-arrestin2 can recruit and subsequently activate Erk1/2 in absence of receptor stimulation. As expected, 5-HT$_4$ receptor-operated Erk1/2 phosphorylation, which is β-arrestin-independent, was not affected by substituting Thr[383] by alanine. This result is consistent with the low Thr[383] phosphorylation level measured in cells expressing 5-HT$_4$ receptors, compared with cells expressing 5-HT$_{2C}$, and suggests that Thr[383] phosphorylation might be a key molecular event governing β-arrestin dependency of Erk1/2 pathway engagement by GPCRs.

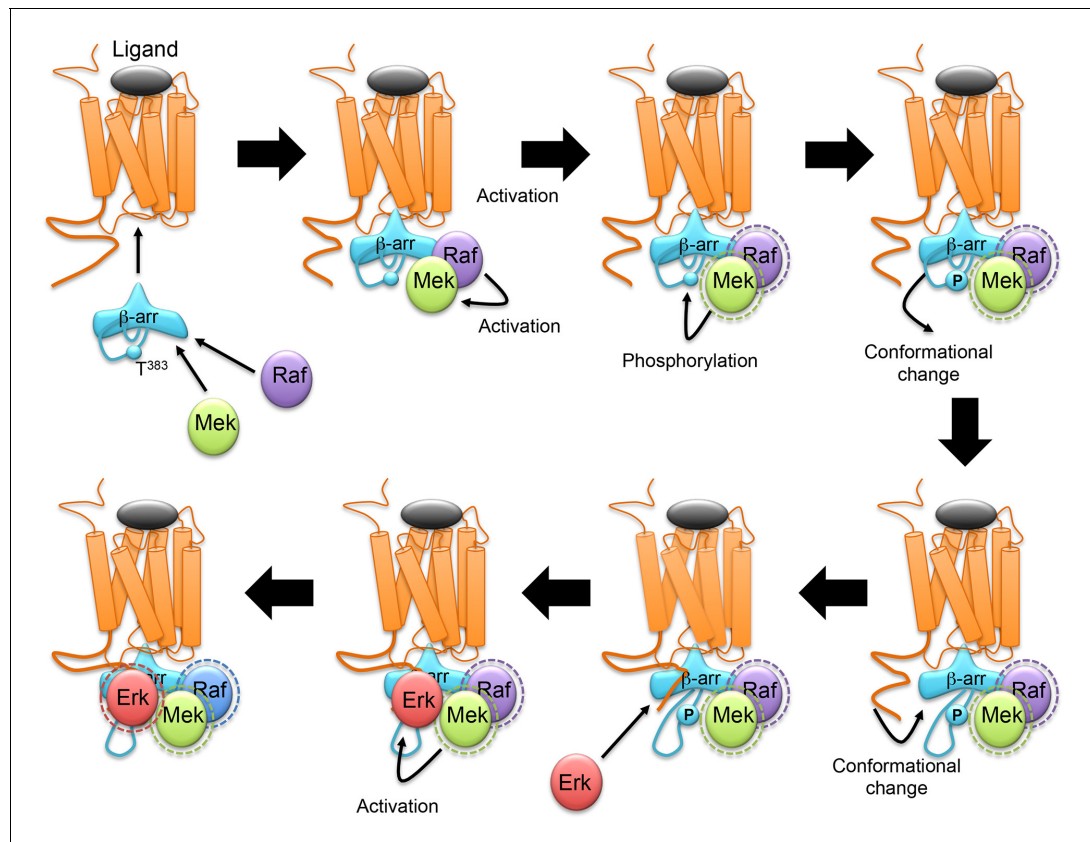

**Figure 6.** Schematic representation of the sequence of events proposed for β-arrestin-dependent Erk1/2 activation by GPCRs. β-arrestin, Raf and MEK are recruited to agonist-stimulated receptor, resulting in MEK activation. Active MEK phosphorylates β-arrestin2 at Thr[383]. This induces a movement of the β-arrestin2 350–393 segment away from the first β-strand of β-arrestin, leaving space for its interaction with the C-terminal domain of the receptor. Erk then binds to the complex and is activated by MEK. The dash circles represent activated enzymes.

In conclusion, our study identifies MEK-elicited $\beta$-arrestin2 phosphorylation at Thr[383] as a key molecular event underlying $\beta$-arrestin-dependent Erk1/2 activation by GPCRs. Thr[383] phosphorylation has recently been involved in $\beta$-arrestin2 association with transient receptor potential vanilloid 1 (TRPV1) at the plasma membrane, which is critical for its role in regulating receptor desensitization (*Por et al., 2013*). This underscores the potentially high influence of Thr[383] phosphorylation upon various $\beta$-arrestin2 functions related or not to GPCRs, while the phosphorylation of other residues (Ser[14] and Thr[276]) might govern steady state level of GPCR cell-surface expression (*Paradis et al., 2015*). This also indicates that $\beta$-arrestin2 can integrate upstream signals not only *via* its conformational changes upon association with receptors but also through a complex phosphorylation pattern that dictates downstream signaling events and the associated physiological functions.

## Materials and methods

### Materials

Human Embryonic Kidney-293 (HEK-293) N type cells (RRID:CVCL_0045) were from the American Type Culture Collection ( anassas, VI, ATCC, CRL-1573). All experiments of the study were performed using a previously characterized batch of cells (Isolate #3) (*Lefkowitz et al., 2002*). Their negative mycoplasma status was analyzed every month using the MycoAlert Mycoplasma Detection Kit (Lonza, Switzerland). Culture media were from Invitrogen (Carlsbad, CA). The siRNA used to target $\beta$-arrestin2 (5'-AAGGACCGCAAAGUGUUUGUG-3', position 201–221) and the control siRNA (5'-AAUUCUCCGAACGUGUCACGU-3') were from Dharmacon (Lafayette, CO)and the GeneSilencer siRNA transfection reagent from Genlantis (San Diego, CA).

pEYFP-$\beta$ARR2, pEYFP-$\beta$ARR1 and pcDNA3-Rluc-$\beta$ARR2 plasmids were kindly provided by Dr. Michel Bouvier (University of Montreal). pcDNA3-Rluc-$\beta$ARR2[T383A] and pcDNA3-Rluc-$\beta$ARR2[T383D] constructs were generated by site directed mutagenesis from the pcDNA3-Rluc-$\beta$ARR2 vector. The pCMV-Rluc8-$\beta$ARR2-RGFP construct used to monitor $\beta$-arrestin2 conformational changes (double brilliance assays), previously described in (*Kamal et al., 2009*), was a generous gift from Dr. Ralph Jockers (Institut Cochin, Paris). pCMV-Rluc8-$\beta$ARR2[T383A]-RGFP and pCMV-Rluc8-$\beta$ARR2[T383D]-RGFP were generated by site directed mutagenesis from this vector. Plasmids encoding Myc-5-HT$_{2C}$-YFP receptor (pEYFP-N1-Myc-5-HT$_{2C}$) and Myc-5-HT$_4$-YFP receptor (pEYFP-N1-Myc-5-HT$_4$) were generated from pRK5-Myc-5-HT$_{2C\text{-}VGV}$ and pRK5-Myc-5-HT$_4$ plasmids (*Chanrion et al., 2008*; *Pellissier et al., 2011*), respectively. The pcDNA3.1-Flag-FSHR plasmid was described in *Tranchant et al. (2011)*. The plasmid encoding catalytically inactive dominant inhibitory mutant of MEK1 (K97A) was a generous gift from Dr. Robert J. Lefkowitz (Duke University Medical Center, Durham, NC).

The mouse anti-phospho-Thr[202]/Tyr[204]-Erk1/2 antibody (Clone E.4, AB_2636855), the rabbit anti-GAPDH antibody (RRID:AB_10167668) and the goat anti-integrin $\beta$1 antibody (RRID:AB_2128200) were from Santa Cruz Biotechnology (Dallas, TX) , the rabbit anti-pan arrestin antibody (RRID:AB_303409) and the mouse anti-$\beta$-arrestin2 antibody (RRID:AB_2060273) from Abcam (UK), the rabbit anti-Erk1/2 antibody (RRID:AB_330744), the rabbit anti-Elk1 (RRID:AB_2277936) and the rabbit anti-phospho-Ser[383]-Elk1 (RRID:AB_2099016) from Cell Signaling Technology (Danvers, MA), the rabbit anti-GFP antibody (RRID:AB_221569) from Invitrogen, and the rabbit anti-Flag (RRID:AB_439687) and mouse anti-Myc (RRID:AB_439694) antibodies from Sigma Aldrich (Saint-Louis, MO). The anti-phosphoThr[383]- $\beta$-arrestin2 antibody was generated by immunizing rabbits with the synthetic CDTNYAT(PO$_3$H$_2$)DDDIVF peptide coupled to Keyhole Limpet Hemocyanin (KLH, Cisbio Bioassays, Codolet, France). Antibodies were purified by two consecutive affinity chromatographies. A first chromatography against the immobilized phosphorylated CDTNYAT(PO$_3$H$_2$)DDDIVF peptide as bait bound antibodies recognizing both phosphorylated and non-phosphorylated forms of the peptide. The retained fraction was then eluted and used for a second chromatography against the immobilized non-phosphorylated CDTNYATDDDIVF peptide as bait. We considered that the non-retained fraction contains antibodies recognizing $\beta$-arrestin2 phosphorylated at Thr[383]. 5-HT, isoproterenol and human recombinant FSH were from Sigma Aldrich, U0126 from Promega (Madison, WI), FR180204 and TBCA from Tocris Bioscience (UK) and CXCL12 from R&D Systems (Minneapolis, MN). [γ-[32]P]-ATP was from Perkin Elmer (Waltham, MA).

## Cell cultures and transfections

HEK-293 cells, grown in Dulbecco's modified Eagle's medium (DMEM) supplemented with 10% dialyzed, heat-inactivated fetal calf serum and antibiotics, were transfected at 40–50% confluence using polyethyleneimine (PEI, Sigma Aldrich), as previously described (*Dubois et al., 2009*) and used 48 hr after transfection. Non-saturating [$^3$H]-mesulergine and [$^3$H]-GR113808 binding experiments showed that 5-HT$_{2C}$ and 5-HT$_4$ receptor densities in transfected HEK-293 cells were 0.29 and 0.90 pmol/mg protein, respectively. Based on previously published levels of the endogenous $\beta$-arrestins in HEK-293 cells (*Ahn et al., 2004b*), we also estimated levels of $\beta$-arrestin1 and $\beta$-arrestin2 as ~1600 pmol/mg protein and ~300 pmol/mg protein in cells transfected with pEYFP-$\beta$ARR1 and pEYFP-$\beta$ARR2 plasmids, respectively, by Western blotting using the rabbit anti-pan arrestin antibody. For experiments using siRNAs, cells were simultaneously transfected with control or $\beta$-arrestin2 siRNA (150 nmol/10$^6$ cells) and plasmids encoding receptors of interest, using the GeneSilencer reagent.

## Quantitative mass spectrometry analysis of β-arrestin1 and β-arrestin2 phosphorylation

HEK-293 cells transiently co-expressing YFP-tagged $\beta$-arrestin1 or YFP-tagged $\beta$-arrestin2 and either Myc-5-HT$_{2C}$ or Myc-5-HT$_4$ receptor were starved of serum for 4 hr, challenged with either vehicle or 5-HT and lysed in 10 mM Tris-HCl, pH 7.4, 150 mM NaCl, 0.5 mM EDTA, 0.5% NP40, 0.4% dodecyl-maltoside, 10 mM sodium fluoride, 1 mM sodium pyrophosphate, 50 mM $\beta$-glycerophosphate, 2 mM sodium orthovanadate and a protease inhibitor cocktail (Roche, Switzerland). Samples were centrifuged at 15,000 × g for 15 min at 4°C and $\beta$-arrestins were immunoprecipitated with agarose-conjugated GFP trap (ChromoTek, Germany). After reduction with dithiothreitol (10 mM, 30 min at 60°C) and alkylation with iodoacetamide (50 mM, 45 min at 25°C), immunoprecipitated $\beta$-arrestins were resolved by SDS-PAGE and digested in-gel overnight at 25°C with trypsin (1 µg, Gold, Promega, Madison, WI). Peptides were analyzed by nano-LC-FT-MS/MS using a LTQ Orbitrap Velos mass spectrometer coupled to an Ultimate 3000 HPLC (Thermo Fisher Scientific, Waltham, MA). Desalting and pre-concentration of samples were performed on-line on a Pepmap precolumn (0.3 mm × 10 mm, Dionex). A gradient consisting of 2–40% buffer B (3–33 min), 40–80% B (33–34 min), 80–0% B (49–50 min), and equilibrated for 20 min in 0% B (50–70 min) was used to elute peptides at 300 nL/min from a Pepmap capillary (0.075 mm × 150 mm) reversed-phase column (Thermo Fisher Scientific). Mass spectra were acquired using a top-10 collision-induced dissociation (CID) data-dependent acquisition (DDA) method. The LTQ-Orbitrap was programmed to perform a Fourier transform (FT) full scan (60,000 resolution) on 400–1,400 Th mass range with the top ten ions from each scan selected for LTQ-MS/MS with multistage activation on the neutral loss of 24.49, 32.66 and 48.99 Th. FT spectra were internally calibrated using a single lock mass (445.1200 Th). Target ion numbers were 500,000 for FT full scan on the Orbitrap and 10,000 MSn on the LTQ. Precursor mass and top 6 per 30 Da windows peak lists were extracted from MS2 using MSconvert 3.0 and searched with Mascot 2.4 (RRID:SCR_014322) against the same human Complete Proteome Set database (RRID:SCR_002380), cysteine carbamidomethylation as a fixed modification and phosphorylation of Ser, Thr and Tyr as variable modifications, 7 ppm precursor mass tolerance, 0.5 Da fragment mass tolerance and trypsin/P digestion. MS2 spectra matching phosphorylated peptides with ion score over 15 were manually inspected for unique transitions that pinpoint the position of phosphorylation sites. Ion signals corresponding to phosphorylated peptides were quantified from the maximal intensities measured in their ion chromatograms manually extracted using Qual browser v2.1 (Thermo Fisher Scientific) with a tolerance of 5 ppm for mass deviation, and normalized to signals of their non-phosphorylated counterparts. For each identified phosphorylated residue, a phosphorylation index (maximal intensity observed in the phosphorylated peptide extracted ion chromatogram/sum of the maximal intensities observed in the phosphorylated and the non-phosphorylated peptide extracted ion chromatograms) was calculated.

## In vitro kinase assays

YFP-tagged $\beta$-arrestin2 (wild type or Thr$^{383}$Ala mutant) was purified from HEK-293 cells using GFP trap beads, as described in the previous section. For kinase assay, 10 µL of purified fractions containing ~1 µg $\beta$-arrestin2 were incubated for 10 min at 37°C in 5 µL kinase buffer (25 mM MOPS, pH 7.2, 12.5 mM $\beta$-glycerophosphate, 25 mM MgCl$_2$, 5 mM EGTA, 2 mM EDTA, 0.25 mM DTT) in

absence or presence of 5 μL active MEK1 (EE, 0.025 μg/μL, Sigma Aldrich). Kinase reaction was started by adding 5 μL ATP (250 μM) and was stopped by adding Laemmli buffer. $\beta$-arrestin2 phosphorylation at Thr[383] was analyzed by Western blotting using the purified anti-phospho-Thr[383]-$\beta$-arrestin2 antibody. For radioactive kinase assay, 10 μl of purified fractions containing ~1 μg YFP-$\beta$-arrestin2 or 1 μg of non-activated, purified GST-tagged Erk2 (Sigma Aldrich) were incubated for 10 min at 37°C in 5 μL kinase buffer in presence of active MEK1 and [γ-$^{32}$P]-ATP (50 μM, 2 μCi/nmol). Proteins were separated by SDS-PAGE and stained with colloidal Coomassie. Gels were dried and exposed for autoradiography. Radioactive bands corresponding to $\beta$-arrestin2 and Erk2 were excised and radioactivity was measured by $\beta$-scintillation counting.

## Western blotting

Proteins, resolved onto 10% polyacrylamide gels, were transferred to Hybond C nitrocellulose membranes (GE Healthcare, UK). Membranes were immunoblotted with primary antibodies (anti phospho-Thr[202]/Tyr[204]-Erk1/2, 1:1000; anti Erk1/2, 1:1000; anti-Flag, 1:1000; Anti-GFP, 1:1000; anti-Myc, 1:1000; anti-phospho-Thr[383]-$\beta$-arrestin2, 1:5000; anti-$\beta$-arrestin2, 1:1000; anti-pan arrestin, 1/1000) and then with either anti-mouse or anti-rabbit horseradish peroxidase-conjugated secondary antibodies (1:5000, GE Healthcare). Immunoreactivity was detected with an enhanced chemiluminescence method (ECL plus detection reagent, Perkin Elmer, Waltham, MA). Immunoreactive bands were quantified by densitometry using the ImageJ software. In protein phosphorylation analyses, the amount of each phosphoprotein was normalized to the corresponding total protein signal.

## ELISA

Quantification of cell surface expression of Myc-tagged 5-HT$_{2C}$ and 5-HT$_4$ receptors was performed by ELISA under permeabilized (Triton X-100 0.05%) vs. non-permeabilized conditions, as previously described (*Chanrion et al., 2008*). Dilutions for primary antibodies were 1:2000 (Anti-Myc) and 1:500 (anti-integrin $\beta$1). Dilutions for secondary antibodies were 1:5000 (donkey anti-goat conjugated to HRP) and 1:8000 (goat anti-mouse conjugated to HRP). Immunoreactive signal was detected with a M200 Infinite plate reader (Tecan), using the SuperSignal ELISA Femto chromogenic substrate (Thermo Scientific). Control experiments were performed using cells transfected with empty vectors and values were normalized with respect to the total amount of protein.

## BRET experiments

HEK-293 cells were transiently co-transfected with either pEYFP-N1-Myc-5-HT$_{2C}$ or pEYFP-N1-Myc-5-HT$_4$ plasmids and either pcDNA3-Rluc-$\beta$ARR2 or pcDNA3-Rluc-$\beta$ARR2$^{T383A}$ or pcDNA3-Rluc-$\beta$ARR2$^{T383D}$ plasmids for $\beta$-arrestin2 recruitment to receptors. For double brilliance $\beta$-arrestin2 assays, they were co-transfected with either pRK5-Myc-5-HT$_{2C}$ or pRK5-Myc-5-HT$_4$ and either pCMV-Rluc8-$\beta$ARR2-RGFP or pCMV-Rluc8-$\beta$ARR2$^{T383A}$-RGFP or pCMV-Rluc8-$\beta$ARR2$^{T383D}$-RGFP. Cells were grown in 96-well micro-plates (6 Í 10$^4$ cells/well) for 48 hr and starved of serum for 6 hr. Cells were washed once with PBS and Coelenterazine H (Sigma-Aldrich) was added at a final concentration of 5 μM for 10 min. Cells were then challenged with 5-HT for 30 min. BRET readings were performed using a Mithras LB940 multimode microplate reader (Berthold Technologies, Germany) allowing sequential integration of luminescence with three filter settings: Rluc filter, 485 ± 10 nm; YFP filter, 530 ± 12.5 nm; RGFP filter, 515 ± 10 nm. Emission signals at 530 nm or 515 nm were divided by emission signals at 485 nm. The difference between this emission ratio obtained with donor and acceptor co-expressed or fused and that obtained with the donor protein expressed alone was defined as the BRET ratio. Results are expressed in milliBRET units corresponding to the BRET ratio multiplied by 1,000.

## Docking

Modeling of the receptor/ß-arrestin/Erk module was made using the PRIOR method, and the PDB structures and methodology indicated in (*Bourquard et al., 2015*). Structural images were made using Pymol.

## Statistics

Statistical analyses were performed using Prism (v. 6.0, GraphPad Software Inc.). Unpaired t test and one-way ANOVA followed by Newman Keuls test were performed to compare two values and for multiple comparisons, respectively.

## Information regarding statistical reporting

For each graph, the number of biological replicates (independent experiments performed on different sets of cultured cells) and significance of observed differences are indicated in the figure legend. The different levels of statistical significance are encoded with symbols (*, **, ***,) directly on the histograms and explained in the figure legends. The statistical tests and software used are indicated in 'Materials and methods' section.

MS peptide identification probabilities were performed using the Mascot algorithm. Mascot search was performed using parameters described in the 'Material and methods' section. Mascot score represents $-10*LOG10$(p-value), where p-value is the absolute probability of peptide wrong assignment.

## Acknowledgements

Funding: This work was supported by grants from la Fondation pour la Recherche Médicale (Contract Equipe FRM 2009), ANR (Contract n° ANR-2011–1619 01), CNRS, INSERM and University of Montpellier to PM and FV, and from ANR (Contract n° ANR-2011–1619 01), INRA and Université de Tours to ER, AP and PC. Mass spectrometry experiments were carried out using facilities of the Functional Proteomic Platform of Montpellier Languedoc-Roussillon, and BRET experiments using facilities of Arpege pharmacological screening platform. The authors wish to thank Cisbio Bioassays for the generation of the anti-phospho-Thr[383]-$\beta$-arrestin2 antibody.

## Additional information

### Funding

| Funder | Grant reference number | Author |
|---|---|---|
| Fondation pour la Recherche Médicale | Contract Equipe FRM 2009 | Elisabeth Cassier<br>Sylvie Claeysen<br>Joël Bockaert<br>Philippe Marin<br>Franck Vandermoere |
| Agence Nationale de la Recherche | Contract ANR-2011-1619 01 | Elisabeth Cassier<br>Nathalie Gallay<br>Thomas Bourquard<br>Sylvie Claeysen<br>Joël Bockaert<br>Pascale Crépieux<br>Anne Poupon<br>Eric Reiter<br>Philippe Marin<br>Franck Vandermoere |
| Centre National de la Recherche Scientifique | | Elisabeth Cassier<br>Sylvie Claeysen<br>Joël Bockaert<br>Philippe Marin<br>Franck Vandermoere |
| Institut National de la Santé et de la Recherche Médicale | | Elisabeth Cassier<br>Sylvie Claeysen<br>Joël Bockaert<br>Philippe Marin<br>Franck Vandermoere |
| Université de Montpellier | | Elisabeth Cassier<br>Sylvie Claeysen<br>Joël Bockaert<br>Philippe Marin |

|  | Franck Vandermoere |
| --- | --- |
| Institut National de la Re-cherche Agronomique | Nathalie Gallay<br>Thomas Bourquard<br>Pascale Crépieux<br>Anne Poupon<br>Eric Reiter |
| Université François-Rabelais | Nathalie Gallay<br>Thomas Bourquard<br>Pascale Crépieux<br>Anne Poupon<br>Eric Reiter |

The funders had no role in study design, data collection and interpretation, or the decision to submit the work for publication.

## Author contributions

EC, performed biochemical and BRET experiments; NG, participated in experiments; TB, AP, performed docking studies; SC, designed and realized molecular tools and participated in biochemical experiments; JB, PC, helped in study conception and manuscript writing; ER, PM, conceived the study, oversaw experiments and wrote the manuscript; FV, conceived experiments, performed LC-MS/MS analyses and some biochemical experiments, oversaw experiments and wrote the manuscript

## Author ORCIDs

Philippe Marin, http://orcid.org/0000-0002-5977-7274

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
