## [Decision Letter]

[Editors’ note: a previous version of this study was rejected after peer review, but the authors submitted for reconsideration. The first decision letter after peer review is shown below.]

Thank you for submitting your work entitled "Phosphorylation of β-arrestin2 at Thr^383^ by MEK underlies β-arrestin-dependent activation of Erk1/2 by GPCRs" for consideration by *eLife*. Your article has been favorably evaluated by Tony Hunter (Senior Editor) and three reviewers, one of whom is a member of our Board of Reviewing Editors. Our decision has been reached after consultation between the reviewers. Based on these discussions, we regret to inform you that we cannot accept your manuscript for publication in *eLife*. We are therefore returning the manuscript to you at this time. Substantial revision will be required to address the criticisms raised during the review. If you are able to fully address these criticisms after the completion of new experimentation, we would happy to consider a new manuscript for publication in *eLife*.

Summary:

This study reports the mechanism by which serotonin 5-HT_2C_ receptors activate the ERK pathway via the β-arrestin2 scaffold protein. The authors propose that MEK promotes the phosphorylation of β-arrestin2 at Thr^383^ and that this is necessary for recruitment of ERK to the receptor complex and its subsequent activation. The findings are potentially very interesting and reveal a novel signaling mechanism for the activation of ERK MAPK downstream of GPCRs. However, further experiments are required to confirm the proposed model and it is a concern that all the experiments rely on the ectopic expression of β-arrestin2 and GPCRs in cells.

Major Points:

1) There is no direct evidence presented that MEK phosphorylates Thr^383^ on β-arrestin2. Does purified MEK phosphorylate β-arrestin2 in vitro? How does MEK phosphorylation of β-arrestin2 compare with that of known MEK substrates?

2) All the experiments, including the MS analysis, are performed on overexpressed YFP-β-arrestin2. It is important to demonstrate that 5-HT induces Thr^383^ phosphorylation of endogenous β-arrestin2 to regulate ERK signaling.

3) The proposed model is that MEK phosphorylation of β-arrestin2 promotes ERK recruitment. To support this, it should be shown that MEK inhibition prevents ERK recruitment to β-arrestin2.

4) CK2 has previously been reported to phosphorylate Thr^383^ on β-arrestin2. The authors suggest in the Discussion that CK2 may be responsible for the basal levels of Thr^383^ phosphorylation. They should demonstrate that inhibition of CK2 does not affect 5-HT-induced Thr^383^ phosphorylation.

5) It is unclear how Thr^383^ phosphorylation causes recruitment of ERK nor do the data presented inform us about the activation mechanism, since increased ERK retention could equally well result in a block in ERK activation.

6) The relative expression of proteins (e.g. receptor and scaffold) in the transfection assays is unclear, and needs to be included.

7) The authors could have provided a more coherent description of the proposed model. The BRET data suggests that the N and C-termini of β-arrestin2 come into closer proximity upon 5-HT treatment and that this is promoted by Thr^383^ phosphorylation (Figure 3). This is consistent with previously reported conformational changes in β-arrestin2 following GPCR activation. However, the model presented in Figure 6, and discussed in the fifth paragraph of the Discussion, describes the C-terminal tail moving away from the receptor/β-arrestin2 complex to create space for ERK recruitment.

[Editors’ note: what now follows is the decision letter after the authors submitted for further consideration.]

Thank you for resubmitting your work entitled "Phosphorylation of β-arrestin2 at Thr^383^ by MEK underlies β-arrestin dependent activation of Erk1/2 by GPCRs" for further consideration at *eLife*. Your revised article has been favorably evaluated by Tony Hunter (Senior Editor) and a Reviewing Editor.

The manuscript has been substantially improved, but there is one remaining issue that needs to be addressed before acceptance, as outlined below:

Point 1: The new data demonstrating that MEK phosphorylates Thr^383^ in β-arrestin2 in vitro are important. However, these data do not address the question in the original decision: "How does MEK phosphorylation of β-arrestin2 compare with that of a known MEK substrate?" A direct comparison of the efficiency of phosphorylation of β-arrestin2 and ERK using an in vitro kinase assay is needed.

---

## [Author Response]

[Editors’ note: the author responses to the first round of peer review follow.]

*Major Points:*

*1) There is no direct evidence presented that MEK phosphorylates Thr^383^ on β-arrestin2. Does purified MEK phosphorylate β-arrestin2* in vitro*? How does MEK phosphorylation of β-arrestin2 compare with that of known MEK substrates?*

Using an in vitro kinase assay followed by Western blotting performed with a newly generated polyclonal antibody against phosphorylated Thr^383^ (see Point 2), we show that purified MEK phosphorylates β-arrestin2 purified from HEK-293 cells on Thr^383^ (see Figure 2 and “Results”, subsection “Role of MEK in β-arrestin2 phosphorylation at Thr^383^ elicited by 5-HT_2C_ receptor stimulation”, last paragraph). As expected, no increase in Thr^383^ phosphorylation was seen in presence of the MEK inhibitor U0126.

*2) All the experiments, including the MS analysis, are performed on overexpressed YFP-β-arrestin2. It is important to demonstrate that 5-HT induces Thr^383^ phosphorylation of endogenous β-arrestin2 to regulate ERK signaling.*

We agree with the reviewers that it is an important issue. Unfortunately, we did not detect by MS/MS any phosphorylated peptide of endogenously expressed arrestins, even after their immunoprecipitation from large cell lysate amounts, likely due to their low expression level in HEK-293 cells and the low immunoprecipitation yield (less than 10%) using commercially available anti-β-arrestin2 antibodies. We thus generated and validated a polyclonal antibody against phosphorylated Thr^383^ (see on Figure 2 the data showing the specificity of the antibody for β-arrestin2 phosphorylated at Thr^383^) and showed that 5-HT_2C_ receptor stimulation, induces a substantial increase in endogenous β-arrestin2 on Thr^383^, while activation of the 5- HT_4_ receptor (known to activate Erk1/2 via a β-arrestin-independent mechanism) does not affect its phosphorylation state, corroborating the results of MS/MS analysis of ectopic β- arrestin2 phosphorylation (see Figure 1 and “Results”, subsection “5-HT_2C_ and 5-HT_4_ receptor stimulation induce distinct patterns of β-arrestin phosphorylation”, last paragraph).

*3) The proposed model is that MEK phosphorylation of β-arrestin2 promotes ERK recruitment. To support this, it should be shown that MEK inhibition prevents ERK recruitment to β-arrestin2.*

Consistent with the proposed model, we show that treating 5-HT_2C_ receptor expressing cells with the MEK inhibitor U0126 abolishes Erk recruitment to β-arrestin2 (assessed by co- immunoprecipitation). This result is illustrated on Figure 4—figure supplement 1 and commented on in the subsection “β-arrestin2 phosphorylation at Thr^383^ is essential for Erk1/2 translocation to the 5-HT_2C_ receptor/β-arrestin2 complex”).

*4) CK2 has previously been reported to phosphorylate Thr^383^ on β-arrestin2. The authors suggest in the Discussion that CK2 may be responsible for the basal levels of Thr^383^ phosphorylation. They should demonstrate that inhibition of CK2 does not affect 5-HT-induced Thr^383^ phosphorylation.*

We show that Thr^383^ phosphorylation elicited upon 5-HT_2C_ receptor stimulation is not affected by treating cells with the selective cell-permeable CK2 pharmacological inhibitor tetrabromocinnamic acid (TBCA, See Figure 2—figure supplement 1, and “Results”, subsection “Role of MEK in β-arrestin2 phosphorylation at Thr^383^ elicited by 5-HT_2C_ receptor stimulation”, first paragraph). Note that previously published studies only reported the phosphorylation of this residue by purified CK2 in vitro. Our results suggest that CK2 plays a minor role (if any) in Thr^383^ phosphorylation in HEK-293 cells and that MEK might be the main kinase responsible for this phosphorylation in living cells. We have thus modified the Discussion accordingly (second paragraph).

*5) It is unclear how Thr^383^ phosphorylation causes recruitment of ERK nor do the data presented inform us about the activation mechanism, since increased ERK retention could equally well result in a block in ERK activation.*

Numerous studies have demonstrated that recruitment of Erk, together with Raf and MEK, to activated GPCR/β-arrestin complex results in Erk activation. In this paper, we propose a mechanistic model whereby β-arrestin2 phosphorylation of Thr^383^ by MEK, within the receptor/β-arrestin2/Raf/MEK complex, is responsible for a large conformational change of β-arrestin2 region comprising residues 350-393. This model is based on (i) the fact that this region in inactive β-arrestin2 interacts with the N-terminal region and has to be displaced for the interaction of this N-terminal region of β-arrestin with the receptor C-tail, (ii) our experimentally validated docking model which shows that in the receptor/β-arrestin/Raf/MEK/Erk complex, the receptor C-tail is sandwiched between β-arrestin and Erk and, consequently, that the conformational change has to take place before interaction with Erk and (iii) our experiments indicating that Erk is not recruited to the complex in absence of this phosphorylation. This is of course only a model. Though different hypotheses could be formulated, this model led us to make the hypothesis and then to validate that MEK is responsible for Thr^383^ phosphorylation, and it accounts for all the existing knowledge. We have largely modified the Discussion (fifth paragraph) to better distinguish the elements that are demonstrated and those which remain hypothetical.

*6) The relative expression of proteins (e.g. receptor and scaffold) in the transfection assays is unclear, and needs to be included.*

We now provide in “Materials and methods” (subsection “Cell cultures and transfections”) the expression levels of 5-HT_2C_ and 5-HT_4_ receptors (0.29 and 0.90 pmol/mg proteins, assessed by non-saturating binding experiments using [^3^H]-mesulergine and [^3^H]-GR113808, respectively) and an estimation (thanks to Western blot experiments) of cellular levels of β-arrestin1 and β-arrestin2 in transfected cells (~1,600 pmol/mg protein and ~300 pmol/mg protein, respectively), based on previously published levels of the endogenous proteins in HEK-293 cells (Ahn et al. J Biol Chem, 279:7807-7811, 2004).

*7) The authors could have provided a more coherent description of the proposed model. The BRET data suggests that the N and C-termini of β-arrestin2 come into closer proximity upon 5-HT treatment and that this is promoted by Thr^383^ phosphorylation (Figure 3). This is consistent with previously reported conformational changes in β-arreston2 following GPCR activation. However, the model presented in Figure 6, and discussed in the fifth paragraph of the Discussion, describes the C-terminal tail moving away from the receptor/β-arrestin2 complex to create space for ERK recruitment.*

The only conclusion that can be raised from our intramolecular BRET experiments is a conformational change of β-arrestin2 upon 5-HT treatment that depends on Thr^383^ phosphorylation. Such experiments do not provide information regarding the exact nature of the conformational change. Our mechanistic hypotheses only concern the region of the β-arrestin2 C-tail comprising residues 350-393, and do not suggest conformational changes affecting the C-terminal extremity. This was indeed unclear in the previously presented model. We have now clarified this point by representing the 393-419 region of β-arrestin2, for which we do not make any hypothesis (see Figure 2 and Figure 6).

[Editors' note: the author responses to the re-review follow.]

*The manuscript has been substantially improved, but there is one remaining issue that needs to be addressed before acceptance, as outlined below:*

*Point 1: The new data demonstrating that MEK phosphorylates Thr^383^ in β-arrestin2 in vitro are important. However, these data do not address the question in the original decision: "How does MEK phosphorylation of β-arrestin2 compare with that of a known MEK substrate?" A direct comparison of the efficiency of phosphorylation of β-arrestin2 and ERK using an in vitro kinase assay is needed.*

As you requested, we now provide an in vitro kinase assay, which shows that MEK1 induces phosphorylation of purified β-arrestin2 and its canonical substrate Erk2 with comparable efficacies. This new result is illustrated on Figure 2—figure supplement 3 and described in the last paragraph of the subsection “Role of MEK in β-arrestin2 phosphorylation at Thr^383^ elicited by 5-HT_2C_ receptor stimulation. The protocol of the radioactive kinase assay used has also been added in the Materials and methods section (subsection “In vitro kinase assays”).